# Differences in Treating Patients with Palpitations at the Primary Healthcare Level Using Telemedical Device Savvy before and during the COVID-19 Pandemic

**DOI:** 10.3390/mi13081176

**Published:** 2022-07-26

**Authors:** Staša Vodička, Erika Zelko

**Affiliations:** Department of Family Medicine, Medical Faculty, University of Maribor, 2000 Maribor, Slovenia; erika.zelko@um.si

**Keywords:** telemedicine, cardiac arrhythmia, primary healthcare level, COVID-19 infection, COVID-19 vaccination, referrals

## Abstract

*Background:* Palpitations are one of most common reasons why a patient visits a general practitioner (GP) and is referred to a cardiologist. Coronavirus disease 2019 (COVID-19) has been associated with new-onset arrhythmias, which are difficult to diagnose at the primary healthcare level during pandemic-related lockdown periods. *Methods*: A total of 151 patients with a complaint of heart rhythm disorder were included from before and during the COVID-19 pandemic, as well as after the start of vaccination, in this cohort retrospective study. We used a telemedical device—namely, a personal electrocardiographic (ECG) sensor called Savvy—to investigate heart rhythm in patients. The primary outcome of the study was to evaluate the number of actual heart rhythm disorder patients and any differences that infection with or vaccination for COVID-19 had on patients handled in a primary healthcare setting. *Results:* We found a heart rhythm disorder in 8.6% of patients before the COVID-19 pandemic and in 15.2–17.9% of patients during the COVID-19 pandemic, where the difference was statistically significant (*p* = 0.002). During the COVID-19 pandemic, we found a heart rhythm disorder in almost 50% of patients that had tested positive for the severe acute respiratory syndrome coronavirus 2 (SARS-CoV-2 virus) more than one month ago. After the vaccinations started, we also found a heart rhythm disorder in almost 50% of non-vaccinated patients. *Conclusions:* Using a telemedical approach or remote consultation is a useful method, at the primary healthcare level, for diagnosing and treating patients with palpitations during times of lockdown.

## 1. Introduction

Palpitations are defined as a sensory symptom accompanied by the unpleasant feeling of a strong, faster, and/or irregular heartbeat, which sometimes can be the only symptom of cardiac arrhythmia [1]. Presenting with palpitations is one of most common reasons why patients visit their GP or their family medicine specialist and are referred to a cardiologist [2]. This represents a significant financial burden to the healthcare system [3], even though actual cardiac arrhythmia is typically discovered in less than half of these patients [4,5]. The prevalence of cardiac arrhythmia in the general population is 2.3% [6], which rises to 12.6% in the elderly population [7]. Differential diagnosis of palpitations includes cardiogenic causes, which are found in 43% of patients, while in 31% the cause is psychogenic; other causes include thyrotoxicosis, caffeine or cocaine consumption, and anemia [8].

These patients are first seen by a GP, where the history of the patient’s current problems, previous medical conditions, and prescribed medications is considered, followed by a physical exam and ECG recordings [9]. Cardiac arrythmias can be sporadic and difficult to diagnose [10]. Therefore, in the case of a normal ECG reading, the patient is usually referred to a cardiologist for further investigation [11]. The following treatment is complex, and expensive and time-consuming tests are performed on patients who do not present any changes in ECG records—mainly to prevent GPs from missing any sporadic arrhythmia in patients with palpitations [12,13,14,15].

Many studies have been conducted on efficiently diagnosing such patients through telemedical approaches or remote consultations [16,17]. Remote consultations can be very useful in general practice because the patient and their GP need not to be at the same place at the same time [18]. This concept was developed in 1980s in order to provide healthcare to patients who live in remote areas [19] and has been shown to be very useful in times of lockdown—such as the COVID-19 pandemic [20]—when patients are not allowed to visit their GP due to restrictions [21] or are not comfortable doing so because of their fear of being infected with the virus [22]. Various low-cost sensors have been used to record the physiological parameters of patients, such as body temperature, heart rate, oxygenation of blood, and blood pressure [16,17,23]. Sensors are connected, by Wi-Fi or Bluetooth, to a smart phone or tablet and can be managed using different applications [24].

COVID-19 has been associated with new-onset arrhythmias that have been difficult to diagnose at the primary healthcare level during lockdown [25]. A German study has reported that 78% patients presented cardiac involvement within two months after COVID-19 diagnosis [26]. Thus, when patients complain to their GP concerning their cardiovascular system in the form of palpitations, a full diagnostic approach must be carried out [27]. Cardiology societies recommend the use of telemedical devices for the diagnosis of patients with palpitations in the context of the COVID-19 pandemic [28].

## 2. Materials and Methods

### 2.1. Study Design

This observational retrospective cohort study is an extension of a previously conducted study at the primary healthcare level [29]. We included patients without previously known heart conditions who had complained about a heart rhythm disorder to their GP. We retrospectively analyzed data collected in three different periods—before the COVID-19 pandemic, during the COVID-19 pandemic, and during the COVID-19 pandemic after vaccination was made available—between August 2019 and October 2021 at a Healthcare Centre in Murska Sobota, Slovenia.

### 2.2. Study Population and Sample Size

Patients aged ≥18 without any previous known cardiac rhythm disorder who visited their GP and complained about a cardiac rhythm disorder or palpitations were included in this study. Patients who had a cardiac rhythm disorder, were severely ill, or were mentally incapable were excluded. We also excluded patients with acute COVID-19 infection or patients previously hospitalized due to severe COVID-19 infection.

To obtain a representative sample, we included 151 patients based on the formula n = P(1 − P) × Zα^2^/d^2^, where n is the calculated sample size, P is the expected proportion in the population (prevalence of cardiac rhythm disorder in the general population is 2.35% [6]), Z is the value of level of confidence (95%), and d is absolute margin of error (0.07).

### 2.3. Data Tool and Collection

We included patients without a previously known heart condition that complained of a heart rhythm disorder or palpitations. However, there had to be no detectable rhythm disturbance on a 12-channel ECG record, which was performed at the first visit before the COVID-19 pandemic. During the COVID-19 pandemic, we performed a psychical exam or remote counseling, according to the COVID-19 restrictions applied at that specific time. If a rhythm disturbance was detected at that time, or if it was highly suspected based on the patient’s history, the patient was treated according to the guidelines and was not included in the study; however, if a rhythm disorder diagnosis could not be made, the patient received a personal ECG sensor (Savvy), which was placed by a healthcare professional and applied onto their chest using two self-adhesive electrodes. The personal ECG sensor Savvy was developed in Slovenia [30] and has been used in a study of 400 patients, which served as the basis for implementing this tool in everyday work by GPs [29]. It is a personal, portable ECG sensor that works with a smartphone using an application called MobECG and a computer program called VisECG, available for Android only. The sensor itself consists of two electrodes placed 8.5 cm apart; it weighs 21 g and has multiple placement positions. The measurement from sensor to smartphone is transmitted by Bluetooth connection. After the measurements are taken, the device is paired with a computer using a USB cable and revised by the VisECG program. We used the necessary personal protection equipment, according to the COVID-19 restrictions applied at that specific time. Then, the sensor was attached to the electrodes and connected by Bluetooth to their smartphone. Patients were instructed to carry out their daily activities as regularly as possible and to keep their smartphones near to them. In case the electrode peeled off, the patient had four spare electrodes that they could install by themself. During the investigation, we asked the patients to keep a journal, where they wrote down their problems and feelings. After three days, the patient returned to the healthcare provider, who removed the sensor and downloaded the measurements from the phone, which were then sent to physicians for analysis. The same patient underwent a checkup with the physician after 5–10 days, either by remote consultation or physical examination.

### 2.4. Data Management and Analysis

The data collected from medical records included baseline demographics, clinical history of certain chronic diseases, and treatments and outcomes of ECG sensor findings. If a cardiac arrhythmia was present, the patient was treated according to medical guidelines and received specific treatment or was referred to a cardiologist or the Emergency department of the local hospital. If a cardiac arrhythmia was not present, the patient was treated according to their symptoms or was kept under observation.

Statistical analyses were conducted using the SPSS^®^ Statistical Program version 26 (SPSS Inc., Chicago, IL, USA) for Windows^®^. For descriptive data, in the case of normally distributed quantitative (numerical) variables, we used the mean ± standard deviation and the minimum and maximum; in case of non-normally distributed quantitative variables, we used the median and the minimum and maximum; for descriptive variables, we used frequencies and percentages.

For the univariable analysis, we used:-*t*-test of independent samples or ANOVA for normally distributed numerical variables;-Mann–Whitney U test for non-normally distributed numerical variables;-Pearson chi-squared test for descriptive variables or Fisher’s exact test as a correction for smaller samples.

A *p*-value ≤ 0.05 was considered to indicate statistical significance in all tests.

### 2.5. Ethical Considerations

The study was approved by the Ethics Committee of the Republic of Slovenia (number 0120-299/2017-7, KME 47/06/17), and was conducted in accordance with The Code of Ethics of the World Medical Association (Declaration of Helsinki). It was also registered on ClinicalTrials.gov (ID: NCT04463524).

Informed consent was obtained from all subjects involved in the study.

## 3. Results

A total of 151 patients complaining about heart rhythm disorder in three different time-frames—before the COVID-19 pandemic (33.0%), during the COVID-19 pandemic (34.5%), and during the COVID-19 pandemic after vaccination (32.5%)—were enrolled.

Table 1 reports the number of patients included in every time frame and the numbers of reported arrhythmias in their time frame alone.

We enrolled 50 patients before the COVID-19 pandemic—13 had arrhythmia; during the COVID-19 pandemic before vaccination, of 50 patients, 28 had arrhythmia; during the COVID-19 pandemic after vaccination, of 49 patients, 25 had arrhythmia.

The main demographic and clinical characteristics of the study population are reported in Table 2.

Patients enrolled in our study were mainly women (74.3%) aged from 19 to 85 years (average, 47.99 years old); 22.5% were smokers and 3.6% were high-risk alcohol consumers. Their BMI ranged from 18 (healthy weight) to 41 (severe obesity) and was 25.77 on average—which corresponds to a normal weight. A total of 37.2% had been diagnosed and treated for arterial hypertension and 42.2% for adult-onset diabetes; 36.3% had been infected with the SARS-CoV-2 virus (in the time frame during COVID-19, 53.8%; in the time frame during COVID-19 after vaccination, 55.1%), and 20.5% had received a vaccine for COVID-19 (in the group during COVID-19 after vaccination, 63.3%). There were no statistically significant differences in the study population in different groups.

### 3.1. Causes for ECG Sensor Placement

The causes for ECG sensor placement in the study population are reported in Table 3. Patients involved in the study mainly reported a feeling of faster heart rate (72.1%), some reported an irregular heart rate (18.4%), and a few reported chest pains (2.4%) or dizziness (6%). There were no statistically significant differences in the study population between different groups.

### 3.2. Heart Rhythm Disorder Diagnosis and Treatment

The presence of actual heart rhythm disorders found in our study group is reported in Table 4, while Table 5 presents the types of heart rhythm disorder found.

We found a heart rhythm disorder in 8.6% of patients before the COVID-19 pandemic and in 16.5–18.5% of patients during the COVID-19 pandemic; this difference was statistically significant (*p* = 0.002; Table 4). During the COVID-19 pandemic, we found a heart rhythm disorder in 13.2% of patients, all of whom had tested positive for SARS-CoV-2 virus more than one month ago. If we only look at their time frame, this means that 20 patients out of 52 had previously had a rhythm disorder (38.5%); after vaccinations started, we also found heart rhythm disorders in 10.5% of patients who had tested positive for SARS-CoV-2 virus more than one month ago and in 5.9% of vaccinated patients. If we only look at their time frame, this means that 25 patients out of 49 had previously had a rhythm disorder (51.0%). The differences between groups were statistically significant (*p* = 0.004).

In 5.2–7.2% of cases, we found supraventricular tachycardiac disorders (atrial fibrillation not included), while in 0–0.7% of cases we observed bradycardic disorders (Table 5). There was no statistically significant difference between these groups. However, there was a statistically significant difference between groups presenting with atrial fibrillation during the COVID-19 pandemic—in 7 out of 52 patients (13.5%; 4.6% of all patients)—before and after the vaccinations started (*p* = 0.004), and in ventricular tachycardiac disorders during the COVID-19 pandemic—in 10 out of 52 patients (19.2%; 6.6% of all patients)—before and after the vaccinations started (*p* = 0.002).

The actions taken for patients at their check-up are reported in Table 6. We found that there was no statistically significant difference in the number of patients treated with observation (17.9–18.5%), that received a specific medication (7.3–9.3%), or were referred to another specialist other than a cardiologist (2.0–3.3%). However, there was a statistically significant difference between the groups in patients referred to a cardiologist (*p* < 0.001). About 35% of patients during the COVID-19 pandemic (in general, 11.2%) were sent to be treated by a cardiologist, compared to 4% before the COVID-19 pandemic.

## 4. Discussion

To the best of our knowledge, this is the first study focused on patients with suspected cardiac arrhythmia at the primary healthcare level using a personal ECG sensor. Palpitations are one of the most common reasons why patients visit their GP and are referred to a cardiologist [2] and can sometimes be the only symptom of an underlying arrhythmia. The primary outcome of our study was to evaluate the number of patients with actual heart rhythm disorders and differences in occurrence before and during the COVID-19 pandemic. We found a statistically significant difference in the discovery of heart rhythm disorders in patients before and during the COVID-19 pandemic (Table 4); during the COVID-19 pandemic, we found heart rhythm disorders in 38.5% of patients that had tested positive for the SARS-CoV-2 virus more than one month ago; after vaccinations started, we also found heart rhythm disorders in 51.0% of unvaccinated patients. The new COVID-19 virus has enhanced the occurrence of cardiac arrhythmia, making it more likely in patients after COVID-19 infection [31]. Researchers have found that cardiac arrhythmia is present in 25–50% of patients [26,32,33], and our study concluded the same for patients before the COVID-19 pandemic (26%). There were no statistically significant differences in the observed groups, which makes our study even more significant, along with the fact that we used a telemedical device—the Savvy ECG sensor—to detect these arrhythmias in a primary healthcare setting.

The acute influences of COVID-19 infection on the wellbeing of patients are diverse, and can be seen in the cardiovascular system, respiratory system, neurological system, and psychological profile [34]. High calorie intake and obesity are important risk factors for developing severe illness after COVID-19 infection [35]. Influences on immune system due to lifestyle habits such as high alcohol intake and smoking also play a role in the severity of the disease [36,37,38]. We found a heart rhythm disorder in almost 50% of patients that had tested positive for SARS-CoV-2 virus more than one month ago; after vaccinations started, we also found a heart rhythm disorder in almost 50% of unvaccinated patients. However, patients before COVID-19 had similar health risks, so we cannot state that we have proven the effects of obesity, diabetes, arterial hypertension, smoking, and alcohol on arrhythmias developed after COVID-19 infection (Table 2). The effects of the pandemic can also be seen at individual and societal levels, as the associated restrictions have led to changes in lifestyle, loss of employment, social distancing, less physical activity, and higher depression and anxiety rates [39].

The exact mechanism of arrhythmogenesis is unknown, but it is probably due to hypoxia caused by pulmonary disease, myocarditis, myocardial ischemia, and electrolytic disbalance in the host immune response [40,41]. The dominant arrhythmia amongst COVID-19 patients is atrial fibrillation [42,43,44,45]. We reported 9.2% more patients (13.5 and 14.3% in their group) suffering from atrial fibrillation in our study who had been infected with the COVID-19 virus in comparison to those before the pandemic (Table 5). There was a higher rate of ventricular tachycardiac disorder in COVID-19 survivors as well (about 6%), similar to other studies performed on patients in hospital settings [46,47].

The patients in our study were treated the same when it came to observation and medicine prescriptions, but there was a significant difference in referrals to cardiologists after COVID-19 infection; this is probably the case as more cardiac arrhythmias were discovered (Table 6). Our actions were like those in other studies, where they mainly observed patients or changed medications in patients without cardiac arrhythmia and referred the patient to a cardiologist if cardiac arrhythmia was detected [48,49,50].

In patients who were vaccinated, we did not find a higher percentage of cardiac arrhythmia than in the group before the COVID-19 pandemic; the literature remains ambiguous in confirming a higher chance of cardiac arrhythmia after vaccination for COVID-19 [51].

Our study is one of few made on patients with cardiac arrhythmia at the primary healthcare level, and the only one involving the usage of the personal ECG sensor Savvy in Slovenia. The patients included in this observational study received the correct treatment at the right time.

There are, however, some limitations to our study. The results were collected in a single healthcare center, without validation in other populations. There was also selections bias, as the population sample was small and collected randomly, consisting of patients who had been treated using a personal ECG sensor (Savvy), and was not stratified by gender, age, or medical conditions present. Due to the small sample size, we could not establish a connection between arrhythmia and COVID-19 infection in this study.

## 5. Conclusions

Patients with palpitations are difficult to assess at the primary healthcare level, especially during times of lockdown due to the COVID-19 pandemic. COVID-19 infection itself carries a higher risk of new-onset cardiac arrhythmia (especially atrial fibrillation), and patients could be left without a diagnosis and proper treatment due to an inability to see their GP in person (e.g., because of the restriction of the fear of being infected with the virus). In our study, we proved that the use of a telemedical approach—namely, the placement of a personal ECG sensor, Savvy—can help in distinguishing an actual heart rhythm disorder from other non-life-threatening causes of palpitations, thus giving patients the opportunity to receive the right treatment at the right time. The use of telemedical devices such as the Savvy ECG sensor could provide a relevant and everyday tool in primary healthcare settings, and its use should be considered in order to enhance the quality of patient treatment and to lower the costs associated with unnecessary referrals to secondary healthcare settings.

## Figures and Tables

**Table 1 micromachines-13-01176-t001:** Patients with presence of arrhythmia in different time frames.

Time Frame	Before COVID-19 Pandemic	During COVID-19 Pandemic	During COVID-19 Pandemic after Vaccination
	*n* (%)	*n* (%)	*n* (%)
All patients included	50 (100)	52 (100)	49 (100)
Arrhythmia present	13 (26)	28 (53.8)	25 (51)
Patients without previous COVID-19 infection	13 (26)	8 (15.3)	NA
Patients with previous COVID-19 infection and unvaccinated	NA	20 (38.5)	16 (32.6)
Patients without previous COVID-19 infection and vaccinated	NA	NA	9 (18.4)

**Table 2 micromachines-13-01176-t002:** Demographic and clinical characteristics of the study population.

Time Frame	Sex		Age	Smoking	Alcohol Consumption		BMI	Arterial Hypertension	Diabetes Type 2	COVID-19 Survivor	COVID-19 Vaccinated
	Male	Female			Moderate	High Risk					
	*n* (%)		Mean ± SD	*n* (%)	*n* (%)		Mean ± SD	*n* (%)	*n* (%)	*n* (%)	*n* (%)
Before COVID-19 pandemic	14 (9.2)	36 (23.8)	47.86 ± 15.514	10 (6.6)	48 (31.7)	2 (1.2)	26.07 ± 5.389	26 (17.2)	23 (15.2)	NA	NA
During COVID-19 pandemic	15 (9.9)	37 (24.6)	49.13 ± 17.112	13 (8.6)	51 (34.3)	1 (0.6)	26.3 ± 4.875	13 (8.8)	25 (16.5)	28 (18.5)	NA
During COVID-19 pandemic after vaccination	10 (6.6)	39 (25.9)	46.98 ± 18.339	11 (7.3)	46 (30.4)	3 (1.8)	24.95 ± 5.451	17 (11.2)	16 (10.5)	27 (17.8)	31 (20.5)
Statistical Significance	NS ^a^		NS ^b^	NS ^a^	NS ^a^		NS ^c^	NS ^a^	NS ^a^	NS ^a^	NA

^a^ χ^2^ test; ^b^ *t*-test for independent samples; ^c^ Mann–Whitney U test; NS, not statistically significant (*p* > 0.05); BMI, body mass index; NA, not applicable.

**Table 3 micromachines-13-01176-t003:** Causes for ECG sensor placement.

Time Frame	Fast Heart Beats	Irregular Heart Beats	Chest Discomfort	Dizziness
	*n* (%)	*n* (%)	*n* (%)	*n* (%)
Before COVID-19 pandemic	36 (23.8)	10 (6.6)	1 (0.6)	3 (1.8)
During COVID-19 pandemic	39 (25.8)	8 (5.2)	2 (1.2)	3 (1.8)
During COVID-19 pandemic after vaccination	34 (22.5)	10 (6.6)	1 (0.6)	4 (2.4)
Statistical Significance	NS ^a^	NS ^a^	NS ^a^	NS ^a^

^a^ χ^2^ test; NS, not statistically significant (*p* > 0.05).

**Table 4 micromachines-13-01176-t004:** Presence of heart rhythm disorder in study population.

Time Frame	Arrhythmia Present in All Patients	Arrhythmia Present in Patients without Previous COVID-19 Infection	Arrhythmia Present in Unvaccinated Patients with Previous COVID-19 Infection	Arrhythmia Present in Vaccinated Patients without Previous COVID-19 Infection
	*n* (%)	*n* (%)	*n* (%)	*n* (%)
Before COVID-19 pandemic	13 (8.6)	13 (8.6)	NA	NA
During COVID-19 pandemic	28 (18.5)	8 (5.2)	20 (13.2)	NA
During COVID-19 pandemic after vaccination	25 (16.5)	NA	16 (10.5)	9 (5.9)
Statistical Significance	*p* = 0.002 ^a^	*p* = 0.004 ^a^	*p* = 0.004 ^a^	NA

^a^ χ^2^ test; NS, not statistically significant (*p* > 0.05); NA, not applicable.

**Table 5 micromachines-13-01176-t005:** Type of heart rhythm disorders present in the study population.

Time Frame	Supraventricular Tachycardiac Disorders Except AF	Atrial Fibrillation	Ventricular Tachycardiac Disorders	Bradycardias
	*n* (%)	*n* (%)	*n* (%)	*n* (%)
Before COVID-19 pandemic	10 (6.7)	0 (0)	2 (1.3)	1 (0.7)
During COVID-19 pandemic	11 (7.2)	7 (4.6)	10 (6.6)	0 (0)
During COVID-19 pandemic after vaccination	8 (5.2)	7 (4.6)	9 (6.0)	1 (0.7)
Statistical Significance	NS ^a^	*p* = 0.004 ^a^	*p* = 0.002 ^a^	NS ^a^

^a^ ANOVA F-test; NS, not statistically significant (*p* > 0.05); AF, atrial fibrillation.

**Table 6 micromachines-13-01176-t006:** Actions taken for patients at check-up.

Time Frame	Observation	Medication(s) Prescribed	Referral to Addition Test (Non-Cardiological)	Referral to Cardiologist
	*n* (%)	*n* (%)	*n* (%)	*n* (%)
Before COVID-19 pandemic	28 (18.5)	13 (8.6)	3 (2.0)	6 (4.0)
During COVID-19 pandemic	28 (18.5)	14 (9.3)	5 (3.3)	15 (9.9)
During COVID-19 pandemic after vaccination	27 (17.9)	11 (7.3)	5 (3.3)	17 (11.2)
Statistical Significance	NS ^a^	NS ^a^	*p* = 0.002 ^a^	*p* < 0.001 ^a^

^a^ ANOVA F-test; NS, not statistically significant (*p* > 0.05).

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
