# Peer review of "Differences in Treating Patients with Palpitations at the Primary Healthcare Level Using Telemedical Device Savvy before and during the COVID-19 Pandemic"

_micromachines, 2022, doi:10.3390/mi13081176_

Round 1

Reviewer 1 Report

This study is interest to primary physicians. However, I think that authors should clarify some points in mauscript.

Before review of this article, I cannot evaluate the quality of English grammar because English is not my mother language. Therefore, this concern should be rechecked by another specialist.

Major 

I think if the main result is emphasized the novelty of this study could be improved.

The primary outcome is to evaluate the number of actual heart rhythm disorder and difference of occurrence according to COVID-19 pandemic. However, the result and conclusion did not clarify this main outcome.

In order to emphasize the main result. I recommend adding “Flow chart” as figure. This could help readers understand the design of this study.

And in the order of result table 1 and 2 is same category as general characteristics, and table 4 is main result of this study. Therefore, I think it is better the table 1 and 2 are merged and then table 4 is placed after general characteristics. 

In discussion, authors mentioned ‘The new Covid-19 virus has enhanced the occurrence of cardiac arrhythmia, making them 196 more likely in patients after Covid-19 infection [30]. Researchers found that cardiac ar-197 rhythmia is present in 25 to 50% of patients [25, 31-32] and our study concluded the same 198 for the patients before the Covid-19 pandemic (22,8%).’ As this citation, the increasement of arrhythmia during COVID-19 is not new finding. Therefore, I think it is more important that these increase arrhythmia patients could be detected by telemedic device and primary physicians. If this finding could be focused in manuscript, the novelty of this study could be improved.

It will be better that in first paragraph of discussion this finding could be more emphasized, and the main findings should be described concisely. 

And this study has many limitations to complaint the conclusion. For scientific reasoning, the limitation of this study should be more discussed and added in limitation section. (For example, selection bias and lack of evidence for relationship between arrhythmia and COVID-19 infection in this study)

I think the conclusion is vague and could not present the main finding of this study. Conclusion should be improved same as above, more concisely and focused on the main finding. 

Minor points

Abstract 

Line 14: “actual hearth rhythm” 

   Please check the spell. In other part you used ‘heart rhythm disorder’

Table 1 and Table 5: Please edit to show the tables in each printed page. In your editing it is hard to read it.

The tables should be promptly cited in results. In result and discussion, there was few citations for tables.

Line 169-173

‘However, there was a statistically significant differences be-168 tween groups in finding an AF during Covid-19 pandemic in almost 14% of patients in their time frame (in general 4.6%) before and after the vaccinations started (p = 0,004) and in finding ventricular tachycardiac disorders during Covid-19 pandemic in almost 20% of patients in their time frame (in general 6.6%) before and after the vaccinations started (p = 0.002).’

To understand this sentence readers should calculate 14% and 20% from table 5. At first I read this I cannot understand the meaning exactly. It is better to rephrasing this sentence or amend the table 5.

Reviewer 2 Report

     The second edition manuscript still needs extensive revision. Therefore, I only recommend this manuscript for publication after major revisions. Several following suggestions may help the further revision:

1)     Abstract format. The current manuscript shows only one paragraph in the abstract. So four titled sections (Background, Methods, Results, and Conclusions) should be removed and revised with more transition terms or sentences. Please check other articles in the Micromachines journal. Or, to keep the same content, adapt a one-paragraph layout for each titled section in the Abstract.

2)     More references and current background information are required. In Line 45 – 55, there is not a sufficient number of references to support the authors' claims: "Many studies", "Various low-cost sensors", and "different application"; only one reference (research article) is cited for these claims. In Line 55 – 57, the personal ECG sensor plays a vital role in collecting data for this study; however, only one sentence is addressed to the ECG sensor. The personal ECG sensors with FDA approval from AliveCor, Inc. have been used in arrhythmia studies even with COVID-19 patients.

3)     Table format. Table 1 (Line 136) and Table 5 (Line 176) are both arranged on two different pages. These tables should show on one page or with extra titled sections on the continued pages. Please keep the table format simple: keep "Fe-male" in one line.

4)     Please keep the statistical values in constancy. In Line 131 – 133, the three different percentages of hearth rhythm disorders are claimed as 33.12%, 34.42%, and 32.45%. If the first value, 33.12% is the combination (33%) of 9.2% and 23.8%, why are the values different in the paragraph and Table 1? Please note how to get these values under the Table and refer to them in the body content.

5)     In Line 139, the authors mention the average 46.80 years old; however, in Table 1, the average ages in three different time frames are 47.86, 49.13, and 46.98. So, how to obtain 46.80 as the average age for 151 patients? Or, if the authors want to emphasize the average for female patients, the average ages in Table 1 should also show the mean values in different gender.

6)     In Line 140, the average BMI, 24.05 is mentioned, but in Table 1, the average BMIs in three different time frames are 26.07, 26.3, and 24.95. So how to get 24.05 as the average BMI?

7)     In Line 150, 73.2% should be corrected as 72.1%, based on Table 3 data.

8)     In Line 157, Table 4 shows 13 (8.6%) values in the column "Arrhythmia present in patients with previous COVID-19 infection". Does it mean before the COVID-19 pandemic, some patients had been confirmed COVID-19 positive? Or this value should be NA, like the last two columns.

9)     In Line 160, please provide precise percentage value (17.9 + 13.2 + 13.2 = 44.3) instead of “almost 50 %”.

10)  In Line 161 & 163, please provide more information about “in general 13.2% & 17.8%”. It should be mentioned the value is from the column of "Arrhythmia present in patients with previous COVID-19 infection and non-vaccinated" in the body context or as a note under Table 4.

11)  In Line 163, please provide precise percentage value (15.2 + 17.8 + 5.9 = 38.9) instead of “almost 50%”.

12)   In Line 166, please use atrial fibrillation instead of AF before Table 5.

13)   In Line 166, 8 – 11% should be corrected as 5.2 – 6.7%, according to Table 5. Besides, in Line 167, 1% should be rectified as 0.7%; in Line 169, 14% should be corrected as 9.2%; in Line 171, 20% should be rectified as 13.9%.

14)  In Line 170, why is 4.6% considered a general value, not 0% (Before the COVID-19 pandemic)? During the COVID-19 pandemic/after vaccination, patients with AF are 4.6% and 4.6%, respectively; there is no change for patients with AF.

15)  In Line 172, why is 6.6% considered a general value? The percentage value of "during the COVID-19 pandemic" dropped from 6.6% to 6% after vaccination. How to get statistical significance based on Fisher's exact test and 6.6% general value? Fisher's exact test is used to determine if there are nonrandom associations between two categorical variables, but there are three categorical variables in the column of AF? 

16)  In Line 179 – 183, please carefully check the statistical values and numbers; most of the numbers do not match Table 6.

17)  In Line 187 – 188, the authors claimed this is the first study in this research field. Unfortunately, this claim conflicts with the manuscript Reference 24 and 25 and lacks literature searches. Please check suggestion 2.

18)  The redundant body content in Line 190 – 195 is the part described in the results in Line 158 – 164. I would suggest combining the Result and Discussion sections.

19)  Please provide accurate values in Line 206 and 208.

20)  In Line 249, please revise the sentence's tone (must). Besides, what does "it" mean here? Personal ECG sensor or Savvy relative product? 

21)  Please keep consistency in reference formatting.

22)  The manuscript needs extensive revision for language and grammar (e.g., Covid-19 or COVID -19)

Round 2
